# Value of ^18^F-PSMA-PET/MRI for Assessment of Recurring Ranula

**DOI:** 10.3390/diagnostics11081462

**Published:** 2021-08-12

**Authors:** Felix Tilsen, Siegmar Reinert, Jürgen Frank Schäfer, Christian la Fougère, Anthony Valentin, Christian Philipp Reinert

**Affiliations:** 1Department of Oral and Maxillofacial Surgery, University Hospital Tuebingen, Eberhard Karls Universität Tübingen, Osianderstrasse 2-8, 72076 Tuebingen, Germany; Siegmar.Reinert@med.uni-tuebingen.de (S.R.); Anthony.Valentin@med.uni-tuebingen.de (A.V.); 2Department of Diagnostic and Interventional Radiology, University Hospital Tuebingen, Eberhard Karls Universität Tübingen, Hoppe-Seyler-Straße 3, 72076 Tuebingen, Germany; Juergen.Schaefer@med.uni-tuebingen.de (J.F.S.); Christian.Reinert@med.uni-tuebingen.de (C.P.R.); 3Department of Radiology, Nuclear Medicine, University Hospital Tuebingen, Eberhard Karls Universität Tübingen, Otfried-Mueller-Straße 14, 72076 Tuebingen, Germany; Christian.LaFougere@med.uni-tuebingen.de

**Keywords:** plunging ranula, MRI, ^18^F-PSMA-PET, PET/MRI, salivary gland, molecular imaging

## Abstract

We report the case of a 6-year-old patient with suspected recurrence of a plunging ranula in clinical and ultrasonographic examination. Surgical resection of the left submandibular and sublingual glands had already been performed. Since persistent glandular tissue could not be excluded with certainty via MRI, we expanded diagnostics by performing a PET/MRI using a head and neck imaging protocol and the radiotracer ^18^F-PSMA-1007, which is physiologically expressed by salivary gland tissue. The ^18^F-PSMA-PET/MRI provided evidence of a cystically transformed, diminishing seroma in the left retro-/submandibular region. No ^18^F-PSMA expressing glandular tissue could be detected in the area of resection, excluding a relapse of a plunging ranula. As a consequence, we opted for a conservative treatment without further surgical intervention. We conclude that a simultaneous ^18^F-PSMA-PET/MRI is a comprehensive imaging modality, which can help to rule out persistent salivary tissue and recurring plunging ranula. It is a useful tool to facilitate the decision making of surgical interventions.

## 1. Introduction

Ranulas are pseudocysts developing from extravasation of mucous after trauma, infection, or obstruction of the sublingual glands and can be classified into simple ranulas, plunging ranulas, and mixed ranulas [1,2].

Simple ranulas are located in the sublingual space (SLS), whereas plunging ranulas are found in the submandibular space (SMS), often communicating to the SLS or parapharyngeal space. Most plunging ranulas are directly passing through the mylohyoidal muscle or through a lateral hiatus remaining from embryonal development. Less frequently, plunging ranulas bypass the muscle from posterior [1].

Surgical treatment options include the excision of the ranula itself and the marsupialization or the excision of the submandibular glad with or without the ranula. While sole excision of the ranula and marsupialization of the gland are less traumatic procedures, their recurrence rate is above 50%. In contrast, the reported recurrence rate after excision of the gland is only 1–2%, albeit a higher surgical trauma and risk of infection, bleeding, nerve lesions, and movement restriction by scar tissue [1]. As a minimally invasive treatment, botulinum toxin type A is described. The toxin is injected into the cyst and the sublingual gland to denervate the parasympathetic fibers and to stop the salivation [3]. Established diagnostic methods are clinical examination, ultrasonography (US), computed tomography (CT), magnetic resonance imaging (MRI), and fine needle aspiration with amylase testing. However, a relevant number of false positive and false negative diagnoses are noted [4]. Due to the rarity of cystic masses in the head and neck area and their nature, they may mimic other cystic lesions [5]. 

Recently, radiolabeled tracers for the visualization of prostate specific membrane antigen (PSMA) by means of positron-emission tomography (PET) have been introduced. PET with ^68^Ga- or ^18^F-labeled PSMA is mainly used as primary imaging tool in patients with prostate cancer and biochemical recurrence [6]. Even at low prostate-specific antigen (PSA) values, PSMA-PET/CT enables a high detection rate of small soft-tissue and bone lesions [7]. Besides the upregulation of PSMA in prostate cancer, normal organs and tissues show physiological PSMA expression [7]. An off-target uptake of PSMA radioligands was reported in glioblastomas, kidneys, small bowels, and lacrimal and major salivary glands [8,9,10].

High physiological uptake of PSMA has been described in healthy salivary, seromucous, and lacrimal glands [10]. The radiotracer uptake is based on the expression of PSMA epitope in glandular cells; therefore, it is hypothesized that the amount of PSMA uptake is associated with the gland volume and functional capacity of the gland [11,12]. 

In head and neck imaging, PSMA-PET/CT has been shown to be beneficial for the individualization of radiotherapy and quantification of salivary gland tissues [10]. Additionally, PSMA-PET/CT may be used for the detection of recurrent and metastatic salivary gland cancers, squamous cell carcinomas or benign salivary gland tumors [13,14]. Whereas anatomical imaging with CT or MRI generally adequately depicts the parotid and submandibular glands, the sublingual glands and minor (mucosal) salivary glands are difficult to visualize. MRI provides better soft-tissue resolution than CT, which is particularly useful to depict small complex masses involving the floor of the mouth or extending through multiple anatomical spaces [15]. With the use of combined PSMA-PET/MRI, the abovementioned relevant imaging information can be obtained in a single examination, thus potentially providing an ideal method for assessment of salivary gland presence and function.

We present a case using PSMA-PET/MRI to exclude persistent salivary tissue and a recurrence of a plunging ranula in a 6-year-old child without further surgery.

## 2. Materials and Methods

### 2.1. ^18^F-PSMA-PET/MRI

Sixty minutes after injection of 45 MBq ^18^F-PSMA-1007, simultaneous PET/MR data were acquired on a 3T PET/MR-system (Biograph mMR, Siemens Healthineers GmbH, Erlangen, Germany), which is able to acquire PET and MRI data simultaneously. The PET acquisition time was 47 min. For the generation of a segmentation-based PET attenuation correction map, a 3D T1-weighted spoiled gradient-echo sequence with Dixon-based fat–water separation was acquired. PET was reconstructed using a 3D ordered-subset expectation–maximization algorithm with 2 iterations, 21 subsets, matrix size 256  ×  256, and Gaussian filtering of 4 mm. The following MRI measurements were performed: T2-weighted transversal and coronal turbo inversion recovery magnitude (TIRM) sequence, T2-weighted sagittal turbo spin echo (TSE) sequence, T1-weighted fast spin echo isotropic 3D sequences, allowing multiplanar reformats, diffusion-weighted imaging (DWI), T1-weighted volumetric interpolated breath–hold examination (VIBE) sequence after intravenous injection of 2 mL gadolinium-based MRI contrast media (GADOVIST^®^, Bayer AG, Leverkusen, Germany).

### 2.2. Image Analysis

Image analysis was performed by two radiologists and a nuclear physician in consensus. The ^18^F-PSMA uptake was quantified by measuring the standardized uptake values (SUVs) and calculation of the SUV mean using 40% isocontour volumes of interests (VOIs).

## 3. Case Report

In 2015, a 6-month-old child presented with a swelling in the anterior floor of the mouth in our clinic. The examination revealed a 20 mm painless swelling in the left floor of the mouth without any restrictions of the tongue mobility. The preliminary diagnosis was a simple ranula. Considering the absence of symptoms, the young age, and possible risks of an anesthesia, we recommended clinical observation. However, within the next six months, the size of the lesion increased. Consequently, we performed an extirpation of the tumor exhibiting a granuloma of the mucosa. Two months afterwards, we had to perform a second surgery. The histopathological diagnosis confirmed a ranula at this time. After performing a marsupialization with good results, there was no evidence of recurrence for four years. 

In November 2019, the patient noticed a painless submandibular swelling on the left side. We performed an ultrasonography showing a 3.5 cm × 2.0 cm × 1.4 cm echo-free cyst and suspected a plunging ranula. This was confirmed by MRI. The patient underwent surgery in June 2020 with extirpation of the ranula and the left sublingual and submandibular glands using a combined intra- and extraoral approach. Shortly after this procedure, we had to drain a postoperative abscess, which showed a normal regression. 

In December 2020, the patient reported pain around the scar accompanied by a soft swelling in the area of surgery. Ultrasonography (Figure 1) revealed an echo-free swelling. Due to postsurgical absence of glandular tissue and scarring, a relapse could be neither diagnosed nor excluded for certain.

The subsequently performed external MRI examination (Figure 2) confirmed a liquid formation in the left submandibular region. However, it was not possible to clearly differentiate between a recurrence of the ranula or a persistent seroma. 

Due to the therapeutic consequences for this patient, we expanded diagnostics performing a PET/MRI using a dedicated head and neck imaging protocol and the radiotracer ^18^F-PSMA-1007. 

^18^F-PSMA-PET/MRI provided evidence of a cystically transformed, size-decreasing seroma in the left retro-/submandibular region (Figure 3). No ^18^F-PSMA-expressing glandular tissue could be detected in the resection area, excluding persistent salivary tissue and a relapse. Thus, a conservative treatment was initiated without further surgical intervention. 

Based on the results of the ^18^F-PSMA-PET/MRI, we recommended further observation of the swelling. During the last visit, the mother reported varying swelling of the left submandibular region without intraoral changes. The patient was pain free, and the tongue movement was only slightly limited. The US showed a diminishing area of low echo density, conforming to a slowly resorbing seroma. Further appointments for checkups were set.

## 4. Discussion

While simple ranulas are limited to the SLS, plunging ranulas are located in the SMS and are treated by an extraoral surgical approach to reduce the rate of recurrence [1]. In our case, the preliminary diagnosis was a simple ranula. Considering the age of the patient, we opted for a low-risk intraoral approach. With this therapeutic strategy, we achieved a temporary remission of the ranula without negative postsurgical changes for four years.

When the patient exhibited renewed signs of a ranula, the child was almost 6 years old and, thus, much more compliant with imaging techniques. To avoid radiation exposure, we used US and MRI, which supported our decision for second surgery. 

At time of the next suspicion of recurrence, US and MRI were performed again. This time, however, a clear differentiation between recurrence and seroma was not possible. Residual glandular tissue could not be ruled out by morphological imaging alone.

As described by Jain et al., conventional imaging methods sometimes prove to be ambiguous or even wrong [4]. Additionally, the reproduction and interpretation of US varies between professionals, which is why in this case only two experienced sonographers performed the US. 

Due to the young age of the patient, an explorative surgery was not justifiable. The use of PSMA-PET/MRI enabled us to clearly exclude residual glandular tissue that would indicated a relapse of a plunging ranula. 

This is in concordance to other studies, showing that minor gland locations can be selectively visualized by a physiologically increased ^18^F-PSMA uptake [10]. Saliva-producing acinar cells are organized in small clusters of mainly mucous cells located in the mucosa of the palate, lips, buccal mucosa, tongue, and floor of the mouth. Anatomical depiction of minor salivary and seromucous glands, which are located in the oral mucosa, lips, tonsils, nasal cavity, larynx, trachea, and esophagus, may only be possible in the case of tumor growth [16]. Although studies suggest that accumulation of PSMA-targeting radioligands in salivary gland tissue is mainly nonspecific [17,18], PSMA PET/CT has been shown to clearly depict normal sublingual and minor submucosal gland areas by using the radionuclide as a marker of the presence of glandular cells [10]. 

The combined acquisition of both morphological information via MRI and metabolic information with PET enables confirmation and differentiation of post-therapeutic anatomical changes in the soft tissue and areas of pathologically increased or decreased function.

The effective PET dose calculated from our described applied activity was approximately 2.5 mSV, which is significantly lower compared to the radiation exposure patients normally receive in PET/CT scans [19,20,21].

Due to the valuable contribution to the diagnosis of ranula, we recommend the use of PSMA-PET/MRI especially for high-risk patients and in case of suspected recurrence.

## 5. Conclusions

The decision of treatment, especially the indication for a surgical approach, has to be carefully considered in high-risk patients with ranula and suspected relapse.

Simultaneous PSMA-PET/MRI is a comprehensive imaging modality to diagnose and clearly exclude persistent salivary tissue in case of recurrence of a plunging ranula. Consequentially, it can prevent unnecessary surgical interventions. 

## Figures and Tables

**Figure 1 diagnostics-11-01462-f001:**
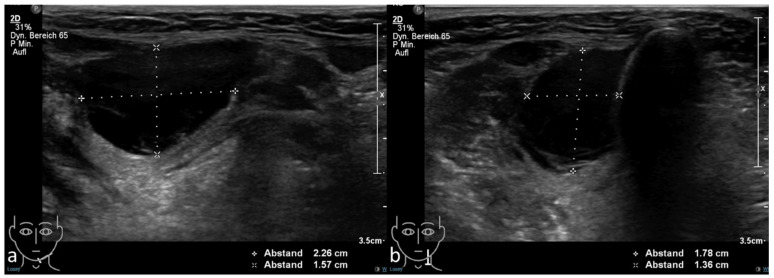
Axial (**a**) and sagittal (**b**) ultrasonography showing an echo-free fluid formation in the left submandibular region.

**Figure 2 diagnostics-11-01462-f002:**
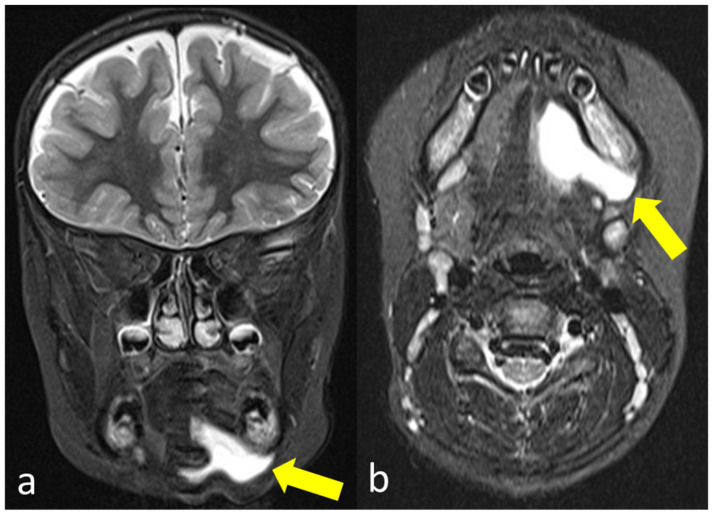
Coronal (**a**) and axial (**b**) MRI with T2-weighted short tau inversion recovery (STIR) sequence showing a hyperintense liquid formation in the left submandibular region (yellow arrows).

**Figure 3 diagnostics-11-01462-f003:**
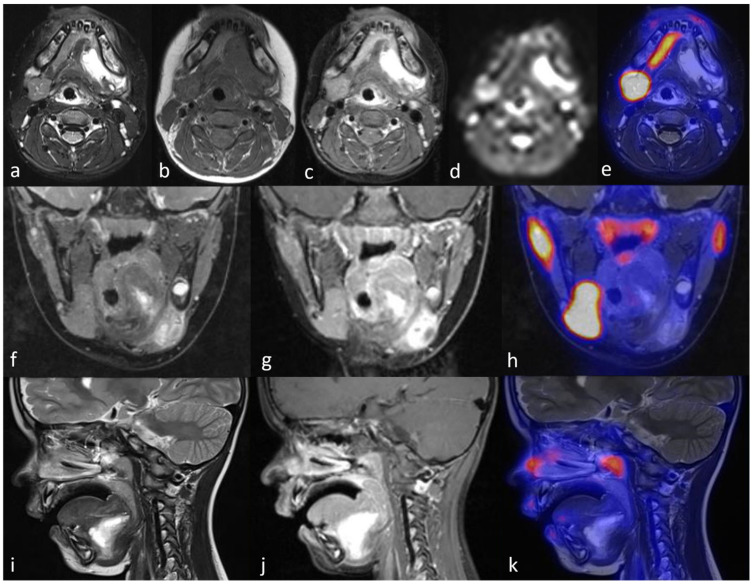
^18^F-PSMA-PET/MRI showing a seroma in the left retro-/submandibular region with hyperintense signal in T2-weighted STIR images (**a**,**f**,**i**), isointense signal in T1-weighted images (**b**), peripherally accentuated enhancement after contrast-medium administration (**c**,**g**,**j**), no diffusion restriction in diffusion-weighed imaging (**d**), and no ^18^F-PSMA uptake in PET (**e**,**h**,**k**).

## Data Availability

The data presented in this study are available on request from the corresponding author. The data are not publicly available due to privacy.

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
