# Peer review of "Value of 18F-PSMA-PET/MRI for Assessment of Recurring Ranula"

_diagnostics, 2021, doi:10.3390/diagnostics11081462_

Round 1

Reviewer 1 Report

The authors report on a pediatric case of suspected recurrence of a ranula. After several courses of surgical treatment conventional diagnostics could not exclude recurrence. Therefore, PSMA-PET was performed and able to exclude recurrence. The authors conclude that PSMA-PET could be used in cases of inconclusive results of suspected salivary gland disease. 

This is an interesting case report and the case is well presented. 

Parts of the template are still present at the end of discussion and introduction and discussion habe a very similar beginning which should be altered. Aside, minor check of spelling/wording (e.g. "operation") should be performed. 

Author Response

Thank you for your constructive feedback on our manuscript.

As recommended, we revised our introduction section and discussion section changing the beginning of the discussion. We further added more literature.

English language and style

( ) Extensive editing of English language and style required
( ) Moderate English changes required
(x) English language and style are fine/minor spell check required
( ) I don't feel qualified to judge about the English language and style

As recommended, a spell check was performed by a native English speaker.

Reviewer 2 Report

lines 124-125 should be deleted.

Interesting case report; however, to be considered for publication, I think that all the discussion section regarding the possible use of 18F-PSMA in salivary gland detection should be enlarged, and an introduction about the marker ( for what it is traditionally used, why you did propose this use, etc...) should be added, as I found very little in medical literature. 

Thank You

Author Response

Thank you for your constructive feedback on our manuscript.

lines 124-125 should be deleted.

We deleted these lines as requested,

Interesting case report; however, to be considered for publication, I think that all the discussion section regarding the possible use of 18F-PSMA in salivary gland detection should be enlarged, and an introduction about the marker ( for what it is traditionally used, why you did propose this use, etc...) should be added, as I found very little in medical literature. 

Thank You

As recommended, we enlarged both the introduction section and the discussion section adding more information about the radiotracer PSMA and citing more relevant literature.

English language and style

( ) Extensive editing of English language and style required
( ) Moderate English changes required
(x) English language and style are fine/minor spell check required
( ) I don't feel qualified to judge about the English language and style

As recommended, a spell check was performed by a native English speaker.

Round 2

Reviewer 2 Report

The paper tremendously improved after revision. The paper is eligible to be published.